**Data Availability Statement:** The dataset is set to be published on 2022-03-01 or after the manuscript is published (with occurs first). Prior to

# Family-based whole-exome sequencing identifies rare variants potentially related to cutaneous melanoma predisposition in Brazilian melanoma-prone families

Felipe Fidalgo[1☯], Giovana Tardin Torrezan[1,2☯], Bianca Costa Soares de Sá[3], Bruna Durães de Figueiredo Barros[1], Luciana Facure Moredo[3], Renan Valieris[4], Sandro J. de Souza[2,5,6], João Pereira Duprat[3], Ana Cristina Victorino Krepischi[7], Dirce Maria Carraro[1,2]*

1 Genomics and Molecular Biology Group, International Research Center/CIPE, A.C.Camargo Cancer Center, São Paulo, Brazil, 2 National Institute of Science and Technology in Oncogenomics and Therapeutic Innovation (INCITO), São Paulo, Brazil, 3 Skin Cancer Department, A.C.Camargo Cancer Center, São Paulo, Brazil, 4 Laboratory of Bioinformatics and Computational Biology, International Research Center, CIPE/A.C. Camargo Cancer Center, São Paulo, Brazil, 5 Bioinformatics Multidisciplinary Environment, Federal University of Rio Grande do Norte, Natal, Brazil, 6 Brain Institute, Federal University of Rio Grande do Norte, Natal, Brazil, 7 Department of Genetics and Evolutionary Biology, Human Genome and Stem-Cell Research Center, Institute of Biosciences, University of São Paulo, São Paulo, Brazil

☯ These authors contributed equally to this work.
* dirce.carraro@accamargo.org.br

## Abstract

Genetic predisposition accounts for nearly 10% of all melanoma cases and has been associated with a dozen moderate- to high-penetrance genes, including *CDKN2A*, *CDK4*, *POT1* and *BAP1*. However, in most melanoma-prone families, the genetic etiology of cancer predisposition remains undetermined. The goal of this study was to identify rare genomic variants associated with cutaneous melanoma susceptibility in melanoma-prone families. Whole-exome sequencing was performed in 2 affected individuals of 5 melanoma-prone families negative for mutations in *CDKN2A* and *CDK4*, the major cutaneous melanoma risk genes. A total of 288 rare coding variants shared by the affected relatives of each family were identified, including 7 loss-of-function variants. By performing *in silico* analyses of gene function, biological pathways, and variant pathogenicity prediction, we underscored the putative role of several genes for melanoma risk, including previously described genes such as *MYO7A* and *WRN*, as well as new putative candidates, such as *SERPINB4*, *HRNR*, and *NOP10*. In conclusion, our data revealed rare germline variants in melanoma-prone families contributing with a novel set of potential candidate genes to be further investigated in future studies.

## 1. Introduction

Ultraviolet radiation exposure is the leading environmental risk factor for the development of cutaneous melanoma [1]. Intermittent sun exposure and sunburns are highly associated with

this, the data can be assessed trough the link below: https://dataview.ncbi.nlm.nih.gov/object/PRJNA705160?reviewer=aqgueip1f84vot7q8qed6br8e2.

**Funding:** GTT, SJS and DMC were funded by Fundação de Amparo à Pesquisa do Estado de São Paulo, grant number 2014/50943-1, Conselho Nacional de Desenvolvimento Científico e Tecnológico, grant number 465682/2014-6 and CAPES - 88887.136405/2017-00. The funders had no role in study design, data collection and analysis, decision to publish, or preparation of the manuscript.

**Competing interests:** The authors have declared that no competing interests exist.

this type of skin cancer [2]. However, hereditary factors play important roles in melanoma etiology, although the genetic basis of melanoma susceptibility is complex and not fully understood [3].

Approximately 10% of all melanoma cases are caused by germline mutations, primarily affecting the p16 isoform of the *CDKN2A* gene, which is responsible for 20–40% of all hereditary melanoma cases [1, 4, 5]. More recently, other genes have been associated with familial melanoma, including *BAP1*, *POT1*, *ACD*, *TERF2IP* and *POLE* [6–8]. Altogether, mutations in these genes, associated with *CDK4*, *TERT* promoter, and *MITF*, are found in < 3% of melanoma-prone families in studied populations, and the majority (>70%) of familial cases are of unknown etiology [4, 8, 9].

Data regarding the prevalence of *CDKN2A* germline mutations in Brazilian patients fulfilling clinical criteria for familial melanoma are scarce; differ by geographic region and adopted diagnostic criteria; and disclose prevalence rates that vary from 4.5% to 14% [10–12]. In our previous study, *CDKN2A* germline mutations were detected in 14% of a cohort of 59 unrelated patients from the Southeast region of Brazil [12]. No *CDK4* pathogenic variants have been identified in Brazilian melanoma-prone families to date [10, 12]. Only one patient with the *MITF* E318K variant was detected in 48 unrelated probands negative for *CDKN2A* variants [13]. Thus, in a significant number of Brazilian melanoma-prone families, no pathogenic variants have been identified, confounding the implementation of adjusted screening and management strategies.

Despite efforts to discover additional melanoma susceptibility genes by using genome-wide approaches such as genome-wide linkage analyses and exome sequencing, studies of either multiple melanoma-affected family members or large case-control cohorts have identified only a small number of candidate loci [14–16]. Thus, the aim of this study was to identify novel genomic variants potentially related to melanoma predisposition in melanoma-prone families. Consequently, we performed whole exome sequencing (WES) of 10 probands from 5 different families (2 probands/family) who developed melanoma and had previously tested negative for *CDKN2A* and *CDK4*.

## 2. Materials and methods

### 2.1. Patients

We selected 5 families with at least 2 cases of cutaneous melanoma among first-degree relatives, for a total of 10 individuals for WES. Selected patients belonged to melanoma-prone families receiving follow-up at the Familial Melanoma Clinic of the Skin Cancer Department and genetic counseling at the Oncogenetics Department at A.C. Camargo Cancer Center (ACC), São Paulo, Brazil. All diagnoses of melanoma were confirmed by histologic review of pathologic materials/reports or medical records. Eligible individuals were those who did not have detectable deleterious mutations in either *CDKN2A* or *CDK4* genes [12]. The ten selected members of the five families were also screened for *TERT* promoter and *MITF* E318K variants [13] and for rare germline copy-number variations [17], with negative results for both analyses, as previously published. This study was conducted in compliance with the Declaration of Helsinki, and was approved by the Internal Ethics Committee Board of A.C.Camargo Cancer Center (#1728/12). All patients provided written informed consent.

### 2.2. Whole exome sequencing

Germline DNA was obtained from peripheral blood leukocytes, following the standard protocols of ACC Biobank. Briefly, DNA was extracted using the Puregene®-DNA purification Kit (Qiagen, Hilden, Germany) according to manufacturer's instructions. DNA concentration,

purity, and integrity were assessed by spectrophotometry (Nanodrop 2000—Thermo Fisher Scientific, Waltham, MA, USA) and fluorometry (Qubit—Life Technologies, Foster City, CA, USA). WES of all 10 patients was performed using the Ion Proton platform (Ion Torrent, Foster City, CA, USA). Genomic libraries were generated with the TargetSeq Exome Enrichment kit (Life Technologies, Foster City, CA, USA) and sequenced on an Ion Proton instrument using Ion PI Sequencing 200 Kit v3 and Ion PI Chip v3 (Thermo Fisher Scientific, Waltham, MA, USA), following the manufacturer's protocol. The resulting sequences were mapped to the reference genome (GRCh37/hg19). Base calling and alignment were performed by using a Torrent Suite v4.2 server and TMAP software (Torrent Mapper 4.2.18). Genomic variant calling was performed in two steps: (1) using the TVC 4.0–5 software (Torrent Variant Caller) following the Ion Torrent protocol (http://mendel.iontorrent.com/ion-docs/); (2) validation of variants using the GATK pipeline (https://www.broadinstitute.org/gatk/guide/best-practices?bpm=DNAseq). The comparison between the number of variants called by each pipeline and the numbers of concordant calls are described in the S1 Table. The exome sequencing data obtained in this study are available at Sequence Read Archive (PRJNA705160).

## 2.3. WES variant prioritization

Variant annotation was performed using public databases: dbNSFP (http://sites.google.com/site/jpopgen/dbNSFP) version 2.4; COSMIC v69; 1000 genomes; Exome Variant Server (http://evs.gs.washington.edu/EVS/) version ESP6500SI-V2; HapMap; and dbSNP version 138 through the SnpEff software version 3.5 using an in-house script developed by the ACC Bioinformatics Department. Variants detected in all 10 samples were disregarded because they could be sequencing artifacts or polymorphisms of the Brazilian population. In addition, our data were compared against an independent set of 20 exomes from Brazilian non-cancer patients (collaboration with the Human Genetics Lab–Dr. Krepischi–Institute of Biosciences, University of São Paulo), and all variants detected in this additional set were also excluded.

Variant prioritization was performed using VarSeq software (Golden Helix), with the following criteria: depth coverage >20 reads; Phred score >20; allelic frequency > 0.2; population frequency <1% (according to NHLBI GO Exome Sequencing Project, the 1000 Genomes Project, the Exome Aggregation Consortium [ExAC], the Online Archive of Brazilian Mutations [ABraOM], and the dbSNP 147). Variants were then selected according to their predicted impact on protein expression: loss of function (frameshift, nonsense, initiator codon alteration, and splice acceptor/donor variants) and missense, including inframe deletions/insertions. Finally, only variants present in both patients from the same family were analyzed further.

## 2.4 Targeted Next-Generation Sequencing (NGS) Validation

A subset of 66 variants selected from exome data were validated by multiplex targeted NGS with a custom Ion AmpliSeq panel. Libraries were prepared with 20 ng of DNA from each patient using an Ion AmpliSeq™ Library Kit 2.0 (Life Technologies), and sequencing was performed using the Ion Proton platform according to the manufacturer's instructions. Sequencing reads mapped to the human genome reference (hg19) using Torrent Suite Browser 4.0.1, and variants were identified using the VariantCaller v4.0.r73742 plugin, considering as criteria for variant calling a base coverage ≥10x and VAF > 20%.

## 2.5. Gene pathway analysis and in silico prediction

We also used gene and pathway analysis software (Ingenuity Pathway Analysis [IPA] and VarElect [http://varelect.genecards.org/]) and *in silico* pathogenicity prediction software to

identify representative pathway networks and to pinpoint other genes that may be important to melanoma susceptibility. The *in silico* pathogenicity prediction scores from SIFT, Polyphen2, LRT, Mutation Taster, Mutation Assessor and FATHMM/MKL software were annotated using VarSeq software (Golden Helix).

## 3. Results

The pedigrees of the 5 investigated families are presented in Fig 1A–1E. Clinical data from all 10 studied individuals are provided in Table 1. The mean age at diagnosis was 40 years old (range 18–65 years old). Half of the patients were diagnosed at age 40 or younger, and most (80%) diagnoses occurred before the sixth decade of life. Six (6/10) patients presented with a single cutaneous melanoma, while 2 cases had multiple lesions (>2). Most patients showed Fitzpatrick phototype I or II (7/10) and nevi counts >50 (6/10). Only 3 cases showed the atypical mole syndrome phenotype. Thyroid cancer and non-melanoma skin cancer where the most prevalent second neoplasms (two cases each) (Table 1).

### 3.1. WES variant prioritization for identifying melanoma predisposing genes

An average of 45,899,246 sequence reads was obtained for each patient and an average of 86% of the target bases was covered more than 20X. First, we used the WES data to investigate variants affecting 10 genes previously associated with melanoma predisposition (*CDKN2A*, *CDK4*, *BAP1*, *POT1*, *TERT*, *ACD*, *TERF2IP*, *POLE*, *MITF*, and *MC1R*), and classified them according to the American College of Medical Genetics and Genomics guidelines [18]. Except for risk alleles in *MC1R*, we did not find any pathogenic/likely pathogenic variants or variants of uncertain significance in these predisposition genes. In three families, one or both relatives

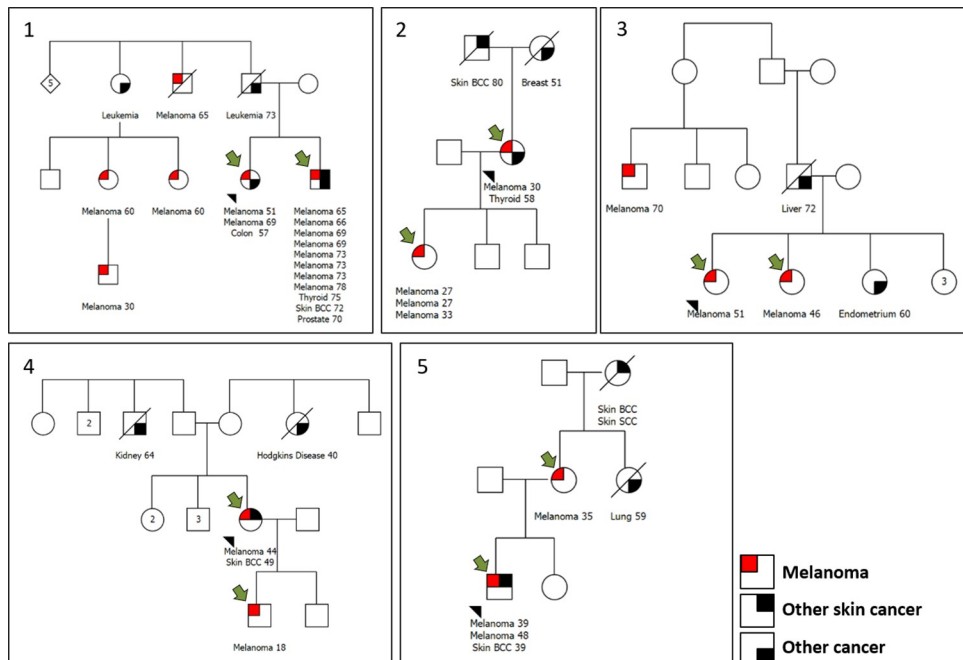

**Fig 1. Pedigrees of the five melanoma-prone families.** Tumor types are described beneath each individual, followed by the age of onset. Small black arrow indicates the index case of each family. The green arrow indicates the individuals who were subjected to WES analysis.

**Table 1. Clinical characteristics of cutaneous melanoma patients.**

| Clinical Aspects | Family 1 | | Family 2 | | Family 3 | | Family 4 | | Family 5 | |
|---|---|---|---|---|---|---|---|---|---|---|
| | A | B | A | B | A | B | A | B | A | B |
| Gender | Female | Male | Female | Female | Female | Female | Female | Male | Male | Female |
| Age at melanoma diagnosis | 51 | 65 | 30 | 27 | 51 | 46 | 44 | 18 | 39 | 35 |
| Number of melanomas | 2 | 8 | 1 | 3 | 1 | 1 | 1 | 1 | 2 | 1 |
| Other cancer | Colon | Prostate Thyroid BCC | Thyroid | - | - | - | BCC | - | BCC | - |
| Kindred | Sister | Brother | Mother | Daughter | Sister | Sister | Mother | Son | Son | Mother |
| Other cancers in family | Melanoma, leukemia | | BCC, breast | | Melanoma, liver, endometrium | | Lymphoma, kidney | | Lung, BCC, SCC | |
| Phototype | I | II | III | III | II | III | II | II | I | I |
| Hair color | Brown | Brown | Brown | Brown | Blond | Brown | Brown | Blond | Blond | Blond |
| Eye color | Brown | Brown | Brown | Brown | Blue | Brown | Brown | Brown | Blue | Blue |
| Nevi count | 100–150 | >150 | >150 | 100–150 | <50 | 50–100 | < 50 | 50–100 | < 50 | < 50 |
| Atypical Mole Syndrome | No | Yes | Yes | No | No | Yes | No | No | No | No |

BCC = basal cell carcinoma; SCC = squamous cell carcinoma of the skin

harbored *MC1R* variants previously associated with increased melanoma risk (low or high-risk variants) (Table 2).

To prioritize variants in other genes, we applied several filters focusing on quality, frequency, and effect of the identified variants and their occurrence in both affected relatives for each family (Fig 2). A total of 288 heterozygous rare non-synonymous variants in 281 genes were identified that co-segregated in both relatives of each family, with 281 missense and 7 loss-of-function (LoF) variants (Table 3 and S2 Table). All variants were exclusive for one given family, and only one gene had different prioritized variants in two families (*UNC93A* gene–Family 1 variant p.Arg226Ter and Family 4 variant p.Gly152Asp). Seven genes harbored rare LoF variants (3 frameshift and 4 nonsense) detected in three families (*ADGRG7*, *FAM221A*, *SERPINB4*, *UNC93A*, *HRNR*, *OR51M1*, *SLC5A11*) (Table 4). We also performed a technical validation of the prioritized variants, selecting a subset of these 288 variants (66 out of 288) for targeted NGS in the same WES samples, and all were validated (S3 Table).

A total of 281 genes were encompassed by the 288 variants. To evaluate the molecular mechanisms and potential roles of these genes in pathogenesis and clinical phenotypes, we performed an analysis using the VarElect tool (http://varelect.genecards.org/), which associates genes and phenotypes based on shared pathways, interaction networks, paralogy relationships,

**Table 2. Clinical characteristics of melanoma patients.**

| Family | Individual | *MC1R* Variants (zygosity) | dbSNP/ ABraOM MAF | Risk classification* |
|---|---|---|---|---|
| 3 | A | p.Val60Leu (ht); p.Arg160Trp (ht) | rs1805005/ 9.8%; rs1805008/ 2.2% | r; R |
| | B | p.Val60Leu (ht) | rs1805005/ 9.8% | r |
| 4 | A | p.Arg160Trp (ht) | rs1805008/ 2.2% | R |
| | B | p.Val60Leu (ht) | rs1805005/ 9.8% | r |
| 5 | A | none | - | - |
| | B | p.Val92Met (ht); p.Thr314 = (ht) | rs2228479/ 3.7%; rs2228478/ 14.5% | r; r |

*R: variants associated with red hair color and more than 2X increased risk for melanoma; r: variants not associated with red color hair and 1-2X increased risk for melanoma [19]. ht: heterozygous. ABraOM: database of Brazilian genomic variants obtained with whole-exome and whole-genome sequencing from 1,171 unrelated individuals (http://abraom.ib.usp.br/index.php). MAF: minor allele frequency.

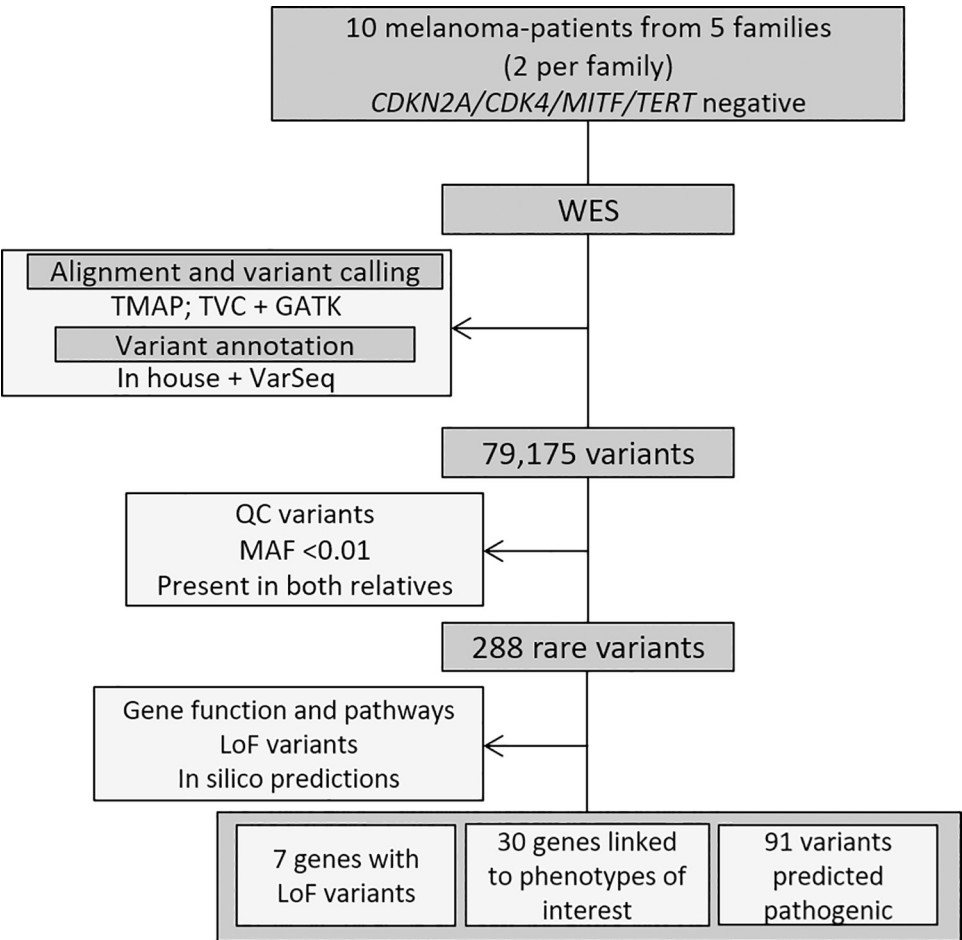

**Fig 2. Diagram of the germline sequencing analysis and variant prioritization strategy, showing the main technical processes.** WES data from 10 melanoma patients were analyzed using quality and frequency-based filters, resulting in 288 rare non-synonymous variants identified in both affected members of each family, which were investigated further for their predicted pathogenicity and gene function. WES, whole exome sequencing. TMAP, Torrent Mapper; TVC, Torrent Variant Caller; GATK, Genomic Analysis Toolkit; QC, Quality Control; MAF, minor allele frequency; LoF, Loss of Function.

and mutual publications. We used the following terms to perform this analysis: cancer; cancer susceptibility; melanoma; melanoma susceptibility; skin pigmentation; melanocyte; melanosome; DNA repair; cell cycle; and telomeres. The most relevant genes (top 20 genes with the highest connection scores) associated with the terms cited above are shown in Table 5; all genes and scores provided by this analysis are provided in S4 Table.

Among the identified rare germline variants, we selected LoF or missense alterations considered deleterious to protein function according to at least four of six prediction algorithms which forecast potential protein malfunctions: SIFT, Polyphen2, LRT, Mutation Taster, Mutation Assessor and FATHMM/MKL. Thus, we obtained a set of 91 rare germline variants affecting 91 different genes and potentially affecting protein function (S5 Table).

### 3.2. Gene pathway analysis

To identify representative pathway networks associated to the 91 genes affected by non-synonymous rare germline variants predicted to affect protein function, we performed an analysis

**Table 3. Types of rare non-synonymous heterozygous variants co-segregating in both affected relatives of the five melanoma-prone families.**

|  | Family 1 | Family 2 | Family 3 | Family 4 | Family 5 | Total |
|---|---|---|---|---|---|---|
| **Total of variants** | 77 | 57 | 72 | 51 | 31 | 288 |
| *Missense* | 73 | 57 | 71 | 49 | 31 | 281 |
| **LoF variants** |  |  |  |  |  |  |
| *Nonsense* | 1 | 0 | 1 | 2 | 0 | 4 |
| *Frameshift* | 3 | 0 | 0 | 0 | 0 | 3 |

LoF: Loss-of-Function

using the Ingenuity Pathway Analysis software (IPA). This set was related principally to cancer, dermatological diseases and conditions, organismal injury and abnormalities, developmental disorders, and hereditary diseases. Furthermore, this classification was associated predominantly with skin cancer, melanoma, and tissue tumorigenesis (p-value <0.00004 –S6 Table).

The main network which comprised 35 genes of the initial set was associated primarily with cancer, organismal injury and abnormalities, and cellular growth and proliferation. These consisted of 13 genes that harbor rare germline variants in melanoma-prone families, and an additional 22 genes that were automatically included in the network because they have been biologically linked to 13 genes implicated by scientific evidence (Fig 3). The category, function/disease, and genes associated to the network are described in Table 6.

## 4. Discussion

Whole genome and whole exome sequencing technologies are powerful tools to identify new cancer predisposition genes. These methods have been applied recently to discover melanoma predisposition genes such as *MITF*, *TERT* and *POT1* [6, 20, 21]. However, despite the description of nearly a dozen melanoma predisposition genes, genetic etiology remains unknown in almost 80% of all melanoma-prone families [9].

In this study, we utilized WES to identify rare germline variants shared between first-degree relatives with cutaneous melanoma from five families, and to discover variants contributing to

**Table 4. Rare germline LoF variants identified by WES in the 5 melanoma-prone families.**

| Family | Gene | dbSNP id | Genomic position (Hg19) | Type | Exon | RefSeq | c. HGVS | p. HGVS | MAF ExAC | MAF ABraOM |
|---|---|---|---|---|---|---|---|---|---|---|
| **1** | *ADGRG7* | rs574492402 | 3:100378552 | Frameshift | 14 | NM_032787 | c.1843_1844insA | p. Pro616Thrfs | 0.0003298 | 0.001642 |
|  | *FAM221A* | rs553824715 | 7:23731209 | Frameshift | 3,4 | NM_199136 | c.631delG | p.Ile212Leufs | n/d | 0.000821 |
|  | *SERPINB4* | rs554627371 | 18:61306960 | Frameshift | 6 | NM_002974 | c.520delC | p. Leu174Trpfs | 0.000008288 | 0.004105 |
|  | *UNC93A* | rs145360877 | 6:167717457 | Stop codon | 4,5 | NM_018974 | c.676C>T | p.Arg226Ter | 0.002356 | 0.004926 |
| **3** | *HRNR* | rs141263661 | 1:152191578 | Stop codon | 3 | NM_001009931 | c.2527C>T | p.Arg843Ter | 0.0002224 | n/d |
| **4** | *OR51M1* | rs182074434 | 11:5410769 | Stop codon | 2 | NM_001004756 | c.141C>G | p.Tyr47Ter | 0.001582 | 0.000821 |
|  | *SLC5A11* | rs147549055 | 16:24873990 | Stop codon | 3,4 | NM_052944 | c.204G>A | p.Trp68Ter | 0.0007087 | n/d |

n/d–not described. MAF–minor allele frequency.

**Table 5. Detected genes associated with the phenotypes of interest and their respective rare non-synonymous variants.**

| Genes | Matched Phenotypes | Matched Phenotypes Count | Score | Average Disease Causing Likelihood (%) | Family | SNP id | HGVS c. | HGVS p. | MAF ExAC |
|---|---|---|---|---|---|---|---|---|---|
| *FANCA* | Cancer, cancer susceptibility, melanoma, skin pigmentation, melanocyte, DNA repair, cell cycle, telomeres | 8 | 164 | 32% | 2 | rs17233141 | c.2574C>G | p.Ser858Arg | 0.01 |
| *WRN* | Cancer, cancer susceptibility, melanoma, skin pigmentation, melanocyte, DNA repair, cell cycle, telomeres | 8 | 146 | 14% | 3 | rs4987238 | c.1149G>T | p. Leu383Phe | 0.001919 |
| *WRN* | Cancer, cancer susceptibility, melanoma, skin pigmentation, melanocyte, DNA repair, cell cycle, telomeres | 8 | 146 | 14% | 3 | rs140768346 | c.2983G>A | p. Ala995Thr | 0.002207 |
| *TYMP* | Cancer, cancer susceptibility, melanoma, skin pigmentation, melanocyte, DNA repair, cell cycle, telomeres | 8 | 70 | 35% | 3 | rs143789597 | c.242G>A | p.Arg81Gln | 0.001112 |
| *NOP10* | Cancer, skin pigmentation, melanocyte, DNA repair, cell cycle, telomeres | 6 | 52 | 66% | 1 | rs146261631 | c.34G>C | p.Asp12His | 0.009744 |
| *PTPN22* | Cancer, cancer susceptibility, melanoma, skin pigmentation, melanocyte, cell cycle, telomeres | 7 | 40 | 29% | 1 | rs72650671 | c.1108C>A | p. His370Asn | 0.002273 |
| *MCM3* | Cancer, cancer susceptibility, melanoma, melanocyte, DNA repair, cell cycle, telomeres | 7 | 35 | 49% | 3 | rs148636199 | c.1618C>T | p. Arg540Trp | 0.00007413 |
| *RECK* | Cancer, cancer susceptibility, melanoma, melanocyte, DNA repair, cell cycle, telomeres | 7 | 33 | 48% | 1 | rs375477269 | c.1747G>A | p.Val583Ile | 0.00004942 |
| *MUC16* | Cancer, cancer susceptibility, melanoma, DNA repair, cell cycle | 5 | 32 | 0% | 1 | rs184811119 | c.14885C>T | p. Thr4962Ile | 0.003101 |
| *MTUS1* | Cancer, cancer susceptibility, melanocyte, DNA repair, cell cycle, telomeres | 6 | 31 | 10% | 4 | rs61733691 | c.1936G>C | p. Glu646Gln | 0.003923 |
| *KMT2D* | Cancer, skin pigmentation, melanocyte, DNA repair, cell cycle | 5 | 26 | 61% | 1 | rs189888707 | c.7670C>T | p. Pro2557Leu | 0.00834 |
| *LRRC56* | Cancer, melanoma, skin pigmentation, melanocyte | 4 | 25 | 15% | 1 | rs61736743 | c.544C>A | p.Gln182Lys | 0.005785 |
| *LRRC56* | Cancer, melanoma, skin pigmentation, melanocyte | 4 | 25 | 15% | 1 | rs138291757 | c.655G>A | p. Val219Met | 0.002642 |
| *HPS5* | Cancer, melanoma, skin pigmentation, melanocyte, DNA repair, cell cycle | 6 | 24 | 16% | 2 | rs143784823 | c.1501G>A | p. Gly501Arg | 0.004406 |
| *ITGA3* | Cancer, cancer susceptibility, melanoma, melanocyte, DNA repair, cell cycle | 6 | 24 | 31% | 1 | rs140248487 | c.2501C>T | p. Thr834Met | 0.0003789 |
| *LCN2* | Cancer, cancer susceptibility, melanoma, melanocyte, cell cycle, telomeres | 6 | 23 | 42% | 5 | rs147787222 | c.26G>T | p.Gly9Val | 0.001367 |
| *ECM1* | Cancer, cancer susceptibility, melanoma, melanocyte, DNA repair, cell cycle | 6 | 23 | 24% | 1 | rs151102225 | c.1181A>T | p. Asp394Val | 0.007734 |
| *DST* | Cancer, melanoma, skin pigmentation, melanocyte, DNA repair, cell cycle, telomeres | 7 | 20 | 50% | 1 | rs138967674 | c.7463C>A | p. Pro2488His | 0.00722 |
| *TTN* | Cancer, melanoma, DNA repair, cell cycle, telomeres | 5 | 20 | 0% | 3 | rs72648244 | c.91573A>G | p. Ile30525Val | 0.00679 |

*(Continued)*

**Table 5.** (Continued)

| Genes | Matched Phenotypes | Matched Phenotypes Count | Score | Average Disease Causing Likelihood (%) | Family | SNP id | HGVS c. | HGVS p. | MAF ExAC |
|-------|-------------------|--------------------------|-------|---------------------------------------|--------|--------|---------|---------|----------|
| *SELP* | Cancer, cancer susceptibility, melanoma, skin pigmentation, melanocyte, DNA repair, cell cycle, telomeres | 8 | 20 | 14% | 1 | rs144853111 | c.2180G>A | p. Gly727Glu | 0.001211 |
| *ADH1B* | Cancer, cancer susceptibility, melanoma, DNA repair, cell cycle, telomeres | 6 | 19 | 20% | 3 | rs6413413 | c.178A>T | p.Thr60Ser | 0.006589 |
| *LAMC1* | Cancer, cancer susceptibility, melanoma, melanocyte, DNA repair, cell cycle | 6 | 18 | 37% | 1 | rs34995260 | c.3796G>A | p. Glu1266Lys | 0.003468 |

MAF–minor allele frequency.

melanoma susceptibility. By applying several filtering strategies, we found 7 LoF variants in three families, and 91 variants predicted to impair gene function by *in silico* analysis. The seven genes affected by LoF variants–*ADGRG7*, *FAM221A*, *SERPINB4*, *UNC93A*, *HRNR*, *OR51M1*, and *SLC5A11* –are not associated with genetic diseases according to the OMIM database and have not been related to melanoma susceptibility previously. Nevertheless, *SERPINB4* and *HRNR* appeared in the "Neoplasia of cells" list identified by the IPA software evaluation of gene pathways of interest.

*SERPINB4* encodes squamous cell carcinoma antigen 2, a member of the serpin family that has serine protease inhibitor functions, and that was initially discovered as a tumor-specific antigen in uterine carcinomas and later described as a biomarker for inflammatory skin diseases [22]. Recently, somatic mutations in *SERPINB4* and *SERPINB3* (predominantly missense mutations) were described in melanoma, and were associated with improved survival after anti-CTLA4 immunotherapy [23]. The second gene, *HRNR*, encodes hornerin, an epidermal protein first described in psoriatic lesions and in cutaneous wound healing [24]. Makino et al. have also shown in a murine model xenotransplanted with human skin that ultraviolet B (UVB) irradiation induces hornerin expression, leading to epidermal hyperproliferation and probably to tissue repair after UVB-induced injury [25]. Hornerin was recently shown to be highly expressed by pancreatic tumor endothelium; to alter tumor vessel parameters through a VEGF-independent mechanism [26]; and to promote tumor progression in human tissues and in cell models of hepatocellular carcinoma [27].

We also used gene and pathway analysis software (VarElect and IPA) and *in silico* pathogenicity prediction software to pinpoint other genes that may be important to melanoma susceptibility. From our list of prioritized variants and genes, three genes (*MYO7A*, *WRN* and *NOP10*) warrant a more detailed discussion, due to gene function and previous associations with melanoma.

A rare missense variant in *MYO7A* was identified in Family 2. Myosin has an essential role in melanosome transport and distribution [28]. In an analysis using IPA software, *MYO7A* was associated with melanoma, cancer, and melanosome degradation and localization. Gibbs et al. found that the absence of *MYO7A* expression in murine retinal pigmented epithelium impaired melanosome motility, thereby impeding the peripheral localization of melanosomes in melanocytes [29], thus showing the importance that *MYO7A* may have on melanocyte homeostasis. Moreover, another *MYO7A* variant (rs2276288) was associated with increased melanoma susceptibility [30].

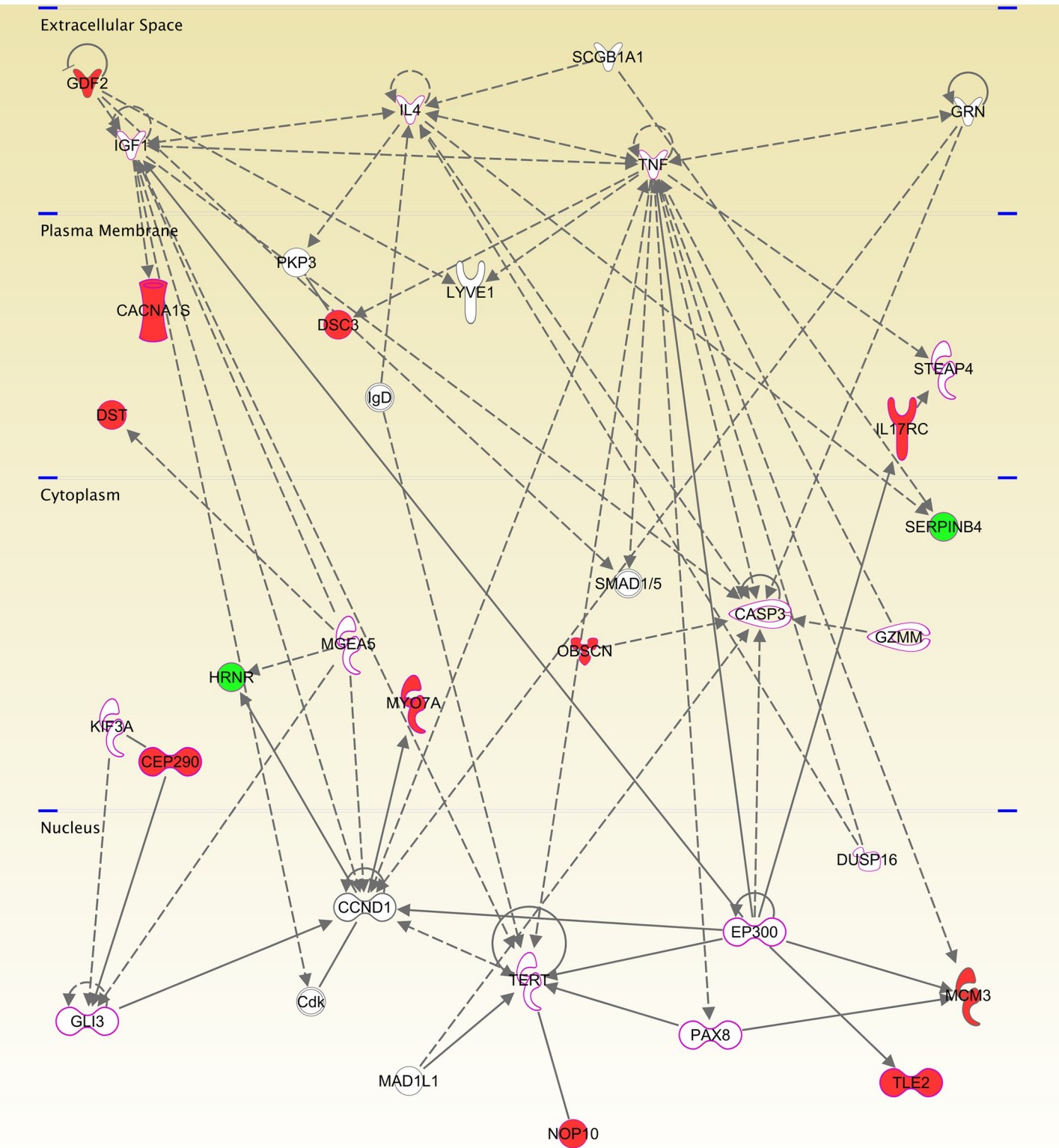

**Fig 3. Network of 13 genes harboring rare germline variants identified in melanoma-prone patients.** Network generated by the IPA software displaying interactions between 13 genes identified by exome sequencing and 22 other genes automatically included after they were identified as biologically connected based on scientific evidence. The functional categorization this network was cancer, dermatological diseases, and organismal injuries and abnormalities. The red nodes represent genes identified by this study harboring missense variants; the green nodes represent genes identified harboring LoF variants; the white node genes were plotted by the software once they are associated by scientific evidence. The nodes highlighted in pink represent genes involved in melanoma tumorigenesis according to IPA.

**Table 6. Main cellular function and diseases associated with 91 prioritized genes.**

| Categories | Diseases or Functions Annotation | Molecules | p-Value |
|---|---|---|---|
| **Cancer, Organismal Injury and Abnormalities** | Connective tissue tumor | *CACNA1S, CCND1, DSC3, DST, GDF2,* ***HRNR,*** *IGF1, IL4, MAD1L1, MCM3, STEAP4, TERT, TLE2, TNF* | 8.07E-08 |
| **Cellular Growth and Proliferation, Tissue Development** | Proliferation of epithelial cells | *CASP3, CCND1, Cdk, EP300, GDF2, GRN, IGF1, IL4, KIF3A, PKP3, STEAP4, TERT, TNF* | 9.46E-08 |
| **Cancer, Organismal Injury and Abnormalities** | Neoplasia of cells | *CACNA1S, CASP3, CCND1, DSC3, DST, EP300, GDF2, GLI3, GRN, GZMM,* ***HRNR,*** *IGF1, IL4, KIF3A, MAD1L1, MGEA5, MYO7A, OBSCN, SCGB1A1,* ***SERPINB4,*** *STEAP4, TERT, TLE2, TNF* | 2.85E-07 |
| **Embryonic Development, Organ Development, Organismal Development, Tissue Development** | Development of sensory organs | *CASP3, CCND1, CEP290, GLI3, GRN, IGF1, IL4, LYVE1, MYO7A, PAX8, TNF* | 5.41E-07 |
| **DNA Replication, Recombination, and Repair** | DNA metabolism | *CASP3, EP300, GDF2, GRN, GZMM, IGF1, IL4, MCM3, MGEA5, TNF* | 5.46E-07 |
| **Cell Morphology, Cellular Function and Maintenance** | Mitochondrial transmembrane potential | *CASP3, CCND1, EP300, GZMM, IL4, MGEA5, TERT, TNF* | 7.09E-07 |

Genes in bold are those with LoF variants.

In Family 3 we identified two rare variants of *WRN*. The *WRN* gene belongs the RecQ subfamily and the DEAH subfamily of DNA and RNA helicases. Consequently, it is associated with DNA transcription, replication, recombination, and repair. Mutants cause the Werner syndrome, an autosomal recessive disorder characterized by progeria and elevated cancer risk. Two variants (rs4733225 and rs13251813) were associated to higher predisposition in melanoma-prone families [31]. Another study of 189 Werner syndrome patients observed that 13.3% developed melanoma, representing a 53-fold elevated risk [32]. Interestingly, one of the variants identified by our study (c.2983G>A; p.Ala995Thr) is contained on the RQC domain, which is responsible for WRN protein-mediated telomere repair [33].

Lastly, we identified a rare variant in the *NOP10* gene (NOP10 Ribonucleoprotein), which interacts directly with *TERT* gene. *NOP10* is a member of the telomerase ribonucleoprotein complex that is responsible for telomere maintenance, thus preserving chromosomal integrity and genome stability [34]. Telomere maintenance genes such as *TERT, ACD, POT1* and *TERF2IP* were associated to melanoma predisposition previously [6–8]. The mutant residue that we found (c.34G>C; p.Asp12His) was described previously in a study of congenital dyskeratosis [35].

We have also compared the 288 genes prioritized in our study with candidate genes reported in nine previous genomic studies of hereditary melanoma [14–16, 36–41] (S7 Table) and only one common gene was identified (*FANCA*). The *FANCA* gene was identified with a suggestive association to melanoma (p = .002) in the TCGA cohort by Yu et al [37]. *FANCA* gene DNA repair gene associated with autosomal recessive Fanconi anemia type A, and there is some preliminary evidence of the association of monoallelic pathogenic variants in *FANCA* and Hereditary Breast and Ovarian Cancer [42] and prostate cancer [43].

We acknowledge that our study has several limitations. First, we only evaluated a small number of families and patients. Second, all of our family duos comprised first degree relatives, which increases the number of shared variants, since any given variant has a 50% chance of being shared between the individuals, do not allow proper linkage analysis and can obscure the identification of pathogenic variants. Third, putative predisposition variants in non-coding or uncaptured regions of the genome (promoter or deep intronic variants) are not detectable by WES.

Also, we did not investigate possible combinatorial effects of more common variants or low penetrance alleles, such as those observed in *MC1R* genes. The *MC1R* gene (melanocortin-1

receptor) is one of the main low/moderate penetrance genes related to cutaneous melanoma. MC1R protein regulates the melanogenesis during exposure to UV radiation and, therefore, has a fundamental role in cutaneous pigmentation [44]. The *MC1R* gene is highly polymorphic, with more than 200 variant alleles been described. Variants called red hair color (RHC) or R alleles are associated with a higher risk (2X) for the development of melanoma as they present loss of receptor function, determining a phenotype of fair skin, ephelides and photosensitivity, in addition to red hair [19, 45, 46]. Non-RHC variants or r alleles determine reduced receptor function and confer less risk (1-2X) for the development of melanoma [19, 45, 47]. In our patients, in two families both relatives harbored *MC1R* variants previously associated with increased melanoma risk (R and/or r alleles) and one family had one relative with two r alleles. The most frequent variant was the r allele p.Val60Leu, which is associated to a 1.47 [19] increased risk of melanoma and was identified in 3 of 10 patients. An R allele (p. Arg160Trp; associated to a 2.69 [19] increased risk of melanoma) was identified in two patients from distinct families.

Finally, although we cannot conclude that any of the identified variants are the definitive cause of melanoma predisposition in these families, our results represent the first WES data from melanoma-prone families in a highly admixed population and provide a set of rare variants with potential roles in melanoma predisposition. The data from our study can contribute to the future identification of genetic similarities between patients evaluated in different studies, facilitating gene discoveries, and furthering the understanding of molecular mechanisms of melanoma carcinogenesis.

## 5. Conclusions

Our data revealed rare germline alterations segregating in patients with familial melanoma, providing new knowledge regarding melanoma predisposition in the Brazilian population. By performing *in silico* analyses of gene function, gene pathways, and variant pathogenicity prediction, we underscored the putative role of particular genes for melanoma risk, contributing with a novel set of potential candidate genes that can be explored further in future studies.

## Supporting information

**S1 Table. TVC X GATK variant calling.**
(XLSX)

**S2 Table. 288 variants prioritized variants.**
(XLSX)

**S3 Table. Targeted NGS validated variants.**
(XLSX)

**S4 Table. VarElect analysis.**
(XLSX)

**S5 Table. Probably pathogenic variants (algorithms).**
(XLSX)

**S6 Table. IPA main diseases and conditions.**
(XLSX)

**S7 Table. Published melanoma candidate genes.**
(XLSX)

## Acknowledgments

We acknowledge the patients who participated in the study, and the A.C. Camargo biobank for sample processing. The authors thank the Coordenação de Aperfeiçoamento de Pessoal de Nível Superior—Brasil (CAPES-Cancer System Biology_88887.136405/2017-00).

## Author Contributions

**Conceptualization:** Ana Cristina Victorino Krepischi, Dirce Maria Carraro.

**Data curation:** Bianca Costa Soares de Sá, Luciana Facure Moredo, João Pereira Duprat.

**Formal analysis:** Felipe Fidalgo, Giovana Tardin Torrezan.

**Funding acquisition:** Sandro J. de Souza, Ana Cristina Victorino Krepischi, Dirce Maria Carraro.

**Methodology:** Felipe Fidalgo, Bruna Durães de Figueiredo Barros.

**Resources:** Sandro J. de Souza, João Pereira Duprat, Ana Cristina Victorino Krepischi, Dirce Maria Carraro.

**Software:** Felipe Fidalgo, Giovana Tardin Torrezan, Renan Valieris.

**Supervision:** Giovana Tardin Torrezan, Ana Cristina Victorino Krepischi, Dirce Maria Carraro.

**Validation:** Felipe Fidalgo.

**Writing – original draft:** Felipe Fidalgo, Giovana Tardin Torrezan, Bianca Costa Soares de Sá.

**Writing – review & editing:** Bruna Durães de Figueiredo Barros, Luciana Facure Moredo, João Pereira Duprat, Ana Cristina Victorino Krepischi, Dirce Maria Carraro.

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
