## [Decision Letter · Decision Letter 0]

20 Apr 2021

PONE-D-21-06922

Family-based whole-exome sequencing identifies rare variants potentially related to cutaneous melanoma predisposition in  Brazilian melanoma-prone families

PLOS ONE

Dear Dr. Carraro,

Thank you for submitting your manuscript to PLOS ONE. After careful consideration, we feel that it has merit but does not fully meet PLOS ONE’s publication criteria as it currently stands. Therefore, we invite you to submit a revised version of the manuscript that addresses the points raised during the review process.

Both reviewers are experts in the field and agreed that the limited sample size does not fully support the conclusions. Among other issues, the reviewers pointed out that gene candidates may be non-significantly associated after a more rigorous statistical analysis, imposing difficulties in interpreting the enrichment analyses. The reviewers also raised the possibility rare founder variants may be missed by excluding the variants detected in all ten samples. Besides, the limitations in this study may have resulted in identifying variants without a relationship with melanoma susceptibility. On the other hand, the reviewers believe there is merit in this study and found it particularly interesting because reports of genetic variants associated with melanoma are scarce for the Brazilian population. Therefore, I invite the authors to address all concerns and comments below, carefully discussing the limitations of the data and avoiding overstatements.

We look forward to receiving your revised manuscript.

Kind regards,

Danillo G Augusto

Academic Editor

PLOS ONE

Journal Requirements:

Reviewers' comments:

Reviewer's Responses to Questions

**Comments to the Author**

1. Is the manuscript technically sound, and do the data support the conclusions?

Reviewer #1: Partly

Reviewer #2: Partly

2. Has the statistical analysis been performed appropriately and rigorously? 

Reviewer #1: No

Reviewer #2: Yes

3. Have the authors made all data underlying the findings in their manuscript fully available?

Reviewer #1: Yes

Reviewer #2: Yes

4. Is the manuscript presented in an intelligible fashion and written in standard English?

Reviewer #1: Yes

Reviewer #2: Yes

5. Review Comments to the Author

Reviewer #1: A study by Fidalgo, Torrezan and colleagues analyzes rare germline variants in families with enriched cutaneous melanoma history. Unfortunately, the number of samples analyzed is small and significantly affects the power of the study, but authors preformed multiple additional analyses to understand the role of identified putative disease genes.

Several questions that need to be addressed to improve understanding of the study and ability to evaluate the results.

Major:

1. It is great that authors tried to validate NGS data and computational pipelines. Some additional information should be included in the manuscript to fully evaluate the validation results.

1.1. As stated in methods (Lines 89-92), Ion Torrent protocol was validated with GATK best practices pipeline. Could authors include a table/figure showing how concordant the results of variant calling with these pipelines were? What are the number of SNPs and Indels per sample?

1.2. Targeted NGS validation is used to validate original NGS sequencing. Conventionally, a short of variants is validated with qPCR/Sanger sequencing – to change both sequencing technology and computational pipeline. Also, coverage > 10X and VAF>20% are incomplete criteria for validation. Were the genotypes the same? It is not clear from the described approach. If so, could you please state the concordance rate?

2. Comparison with non-cancer exomes, used as one of the filtration steps in variant prioritization should be described in more details. Were the samples jointly called? How was systematic difference between case and control dataset assessed (e.g. using number and frequencies of common synonymous variants?

3. Formal statistical testing for significance of co-segregation is missing. It would be great to have any test (e.g. LOD score) performed to show how significant is co-segregation of identified variants with the phenotype, especially, given a correction for multiple test hypothesis. All the downstream pathway analyses relies on co-segregation which was not formally assessed for significance, therefore, is hard to reliably interpret.

Minor:

link [39] is out of order on line 73.

Reviewer #2: The authors conducted whole exome sequencing on 10 affected individuals from 5 melanoma-prone familiesThe authors conducted whole exome sequencing on 10 affected individuals from 5 melanoma-prone families negative for mutations in CDKN2A and CDK4. The evaluations revealed 288 rare co-segregating coding variants across the 5 families. The authors conducted in silico, gene and pathway-based evaluations to further prioritize variants/genes for further follow-up. Based on these additional evaluations and review of the literature, the authors prioritize several potential candidate genes for further study.

Abstract. Given the small sample size in this study, the authors should modify the conclusions to indicate that the proposed genes are potential candidates. With the current study, it is not possible to conclude which genes are true candidates for melanoma genetic susceptibility.

Materials and Methods. The authors excluded variants if they were detected in all 10 samples. Although such variants would likely be sequencing artifacts or polymorphisms, there is the remote possibility that the authors might have detected a rare founder variant in their population that was responsible for disease in all 5 families. Did the authors evaluate any of the population-level rare variants observed in all 10 samples to make certain that they did not potentially reflect disease-related founder variants?

Depth coverage>50 was used as a filtering criteria. However, the authors report that the sequencing was conducted such that an average of 86% of the target bases were covered more than 20X. What proportion of variants were thus excluded based on depth coverage > 50 being required for retention of variants for further study?

Results. The authors technically validated a subset of the 288 prioritized variants. Were any family members available for further co-segregation evaluation of the prioritized variants in any of the 5 sequenced families?

Discussion. The authors acknowledge the limitations related to the small sample size of their study. Did the authors interrogate publicly available archives/databases to search for data from other melanoma cases and families to further investigate the prioritized variants/genes identified in this study?

6. PLOS authors have the option to publish the peer review history of their article (what does this mean?). If published, this will include your full peer review and any attached files.

Reviewer #1: No

Reviewer #2: No

---

## [Author Response · Author response to Decision Letter 0]

21 Jun 2021

To Prof. Dr. Danillo G Augusto

Academic Editor, PlosOne

 We are submitting a revised version of the manuscript “PONE-D-21-06922” for your reconsideration. The manuscript entitled “Family-based whole-exome sequencing identifies rare variants potentially related to cutaneous melanoma predisposition in Brazilian melanoma-prone families” was reviewed according to the suggestions made by the reviewers. 

 Below we present the point-to-point response to each comment and suggestion. All authors agree with the final version of the manuscript.

 I hope you find it appropriate for publishing in PlosOne as a Research Article. 

Very best wishes,

Dirce Maria Carraro

Genomics and Molecular Biology Group, International Research Center

A.C. Camargo Cancer Center, R. Taguá, 440, Liberdade, São Paulo – SP, Brazil.

e-mail: dirce.carraro@accamargo.org.br

Reviewer #1: A study by Fidalgo, Torrezan and colleagues analyzes rare germline variants in families with enriched cutaneous melanoma history. Unfortunately, the number of samples analyzed is small and significantly affects the power of the study, but authors preformed multiple additional analyses to understand the role of identified putative disease genes.

Several questions that need to be addressed to improve understanding of the study and ability to evaluate the results.

Response: We thank reviewer one for the careful assessment of our manuscript. We understand the concern of the reviewer regarding the small number of patients included in the study. However, we would like to clarify that by the time of patients’ selection, we have attempt to contact relatives from other melanoma-prone families from our cancer registry (which included 50 individuals with clinical criteria for hereditary melanoma screened for mutations in the main associated genes), and due to the difficulty of including alive affected relatives, we were successful in including only the 10 patients from 5 families that are described in our manuscript. Also, we need to clarify that the funding we had available for performing this project was limited, which was impeditive for performing WES in more individuals, and was the reason why we have opted to focus on a family-based study of highly selected individuals. During the past months, we have tried to recruit other melanoma affected family members for 2 of the 5 families from our study, in other to improve the variant prioritization of these families; however, we were not able to recruit any additional relatives. 

 Although we understand the limitations that this small number of patients impose to our findings, our results represent the first WES data from melanoma-prone families in Brazil, a highly admixed population with scarce published genomic data. The identified set of rare variants with putative roles in melanoma predisposition can contribute to the future identification of genetic similarities between patients evaluated in different studies, advancing the understanding of molecular mechanisms of melanoma carcinogenesis. 

Major:

1. It is great that authors tried to validate NGS data and computational pipelines. Some additional information should be included in the manuscript to fully evaluate the validation results.

Response: We included a more detailed information regarding the validation of prioritized variants, as described in detail in question 1.2. 

1.1. As stated in methods (Lines 89-92), Ion Torrent protocol was validated with GATK best practices pipeline. Could authors include a table/figure showing how concordant the results of variant calling with these pipelines were? What are the number of SNPs and Indels per sample?

Response: We have included a supplementary table (S1) containing the total number of variants called by each pipeline, as well as the concordant number of variants and unique number of variants. Briefly, the mean number of SNPs called by TVC was 33,563 and by GATK was 26,747, with a concordance mean of 26,123 SNPs. For indels, the mean number of called variants was 1,654 for TVC and 987 for GATK, with a mean concordance of 741. 

1.2. Targeted NGS validation is used to validate original NGS sequencing. Conventionally, a short of variants is validated with qPCR/Sanger sequencing – to change both sequencing technology and computational pipeline. Also, coverage > 10X and VAF>20% are incomplete criteria for validation. Were the genotypes the same? It is not clear from the described approach. If so, could you please state the concordance rate?

Response: We corrected this information in the manuscript and included more detailed data regarding the validation of prioritized variants (Methods section, line 119; Result section line 168). The complete list of validate variants, their coverage, allele frequency and zygosity in both the WES and the targeted NGS sequencing is now presented in the supplementary table S3. We would like to clarify that although the criteria of coverage >10X and VAF >20% were used for setting the variant caller software, our validation data had coverage and frequencies much higher. The mean coverage of the validated variants was 16,217 (range 32 – 147,579) and the mean variant allele frequency was 0.5 (0.42 – 0.61). All variants were confirmed as heterozygous, as expected from WES data. When considering the number of candidate variants that were validated, the correct number is 66 from the 288 candidate variants (in the previous version of the manuscript we mentioned 79 variants, but 13 of these variants were excluded from our final 288 candidate variants list after being validated, due to new prioritization criteria).

2. Comparison with non-cancer exomes, used as one of the filtration steps in variant prioritization should be described in more details. Were the samples jointly called? How was systematic difference between case and control dataset assessed (e.g. using number and frequencies of common synonymous variants?

Response: The non-cancer exome samples were called individually and were provided from another study of one co-author. As in the time of our initial analysis and variant selection (performed in 2016) the ABraOM database was not available (which now contains exome sequencing data for 1,171 Brazilian healthy individuals), these 20 non-cancer individuals were used only to exclude variants that could be common in the Brazilian population and absent in another genomic database. For this filter, we have only received a VCF file containing the variants from the non-cancer samples and compared with variants from our patients. No additional comparison was performed between these cohorts.

3. Formal statistical testing for significance of co-segregation is missing. It would be great to have any test (e.g. LOD score) performed to show how significant is co-segregation of identified variants with the phenotype, especially, given a correction for multiple test hypothesis. All the downstream pathway analyses rely on co-segregation which was not formally assessed for significance, therefore, is hard to reliably interpret.

Response: All our family duos consist of first-degree relatives, which leads to a percentage of shared variants of 50% by chance. To be able to perform a linkage analysis with significance and obtain significant LOD scores several members of the same family, with different degrees of kinship, should be evaluated. This is a recognized limitation of the study and is now better described in the discussion section of the manuscript (line 81): “Second, all of our family duos comprised first degree relatives, which increases the number of shared variants, since any given variant has a 50% chance of being shared between the individuals, do not allow proper linkage analysis and can obscure the identification of pathogenic variants”. 

Minor:

link [39] is out of order on line 73.

Response: Literature citations order was corrected in the manuscript.

Reviewer #2: The authors conducted whole exome sequencing on 10 affected individuals from 5 melanoma-prone families. The authors conducted whole exome sequencing on 10 affected individuals from 5 melanoma-prone families negative for mutations in CDKN2A and CDK4. The evaluations revealed 288 rare co-segregating coding variants across the 5 families. The authors conducted in silico, gene and pathway-based evaluations to further prioritize variants/genes for further follow-up. Based on these additional evaluations and review of the literature, the authors prioritize several potential candidate genes for further study.

Response: We thank reviewer two for the careful assessment of our manuscript. We have addressed the reviewer questions below.

Abstract. Given the small sample size in this study, the authors should modify the conclusions to indicate that the proposed genes are potential candidates. With the current study, it is not possible to conclude which genes are true candidates for melanoma genetic susceptibility.

Response: We agree with the reviewer that the identified genes are potential candidates. We have modified the abstract and parts of the discussion and conclusion to avoid overstatement of our findings.

Materials and Methods. The authors excluded variants if they were detected in all 10 samples. Although such variants would likely be sequencing artifacts or polymorphisms, there is the remote possibility that the authors might have detected a rare founder variant in their population that was responsible for disease in all 5 families. Did the authors evaluate any of the population-level rare variants observed in all 10 samples to make certain that they did not potentially reflect disease-related founder variants?

Response: We did not evaluate the presence of any of the population-level rare variants observed in all 10 samples in our first analyses. To answer the request of reviewer 2, we had re-analyzed our data and identified 5,085 variants that were present in all 10 patients. After excluding variants with MAF >0.5% in ExAc and ESP, classified as benign in Clinvar and non-coding variants (except splice site variants), there were 49 variants present in all 10 patients. 39 variants were in 2 genes that presented bad alignment in the regions of the variants. The remaining 10 variants were visually inspected, and 9 were most likely sequencing artifacts while one was most likely a variant present in one of the pseudogenes from CDC27 gene.

Depth coverage>50 was used as a filtering criteria. However, the authors report that the sequencing was conducted such that an average of 86% of the target bases were covered more than 20X. What proportion of variants were thus excluded based on depth coverage > 50 being required for retention of variants for further study?

Response: We apologized for the incorrect value in the first version of the manuscript, we corrected the coverage for 20X (methods: line 106).

Results. The authors technically validated a subset of the 288 prioritized variants. Were any family members available for further co-segregation evaluation of the prioritized variants in any of the 5 sequenced families?

Response: We have tried extensively to recruit other melanoma affected family members for 2 of the 5 families from our study that had more melanoma patients, in other to improve the variant prioritization of these families. However, unfortunately we were not able to recruit any additional affected relatives. We were able to recruit one unaffected member of one family; however, as the underlying genetic causes of melanoma predisposition in these families could be related to genes with moderate or low penetrance, it would be of limited relevance to perform the co-segregation of identified variants in unaffected family members.

Discussion. The authors acknowledge the limitations related to the small sample size of their study. Did the authors interrogate publicly available archives/databases to search for data from other melanoma cases and families to further investigate the prioritized variants/genes identified in this study?

Response: We did not analyze raw data from other melanoma studies. However, we performed an extensive literature review to identify other studies of that investigated candidate genes to melanoma predisposition. We have identified 9 studies that used comprehensive genomic approaches (WES and WGS) in melanoma patients and compared the candidate genes found in these studies (supplementary table S7) with the list of genes prioritized in our study. From our 288 initial candidates, only one the FANCA was identified as a candidate gene to melanoma risk in other study (Yu et al, 2018). A paragraph regarding these comparisons was added to the discission section (lines 73-78).

---

## [Decision Letter · Decision Letter 1]

11 Jul 2021

PONE-D-21-06922R1

Family-based whole-exome sequencing identifies rare variants potentially related to cutaneous melanoma predisposition in  Brazilian melanoma-prone families

PLOS ONE

Dear Dr. Carraro,

Thank you for submitting your manuscript to PLOS ONE. After careful consideration, we feel that it has merit but does not fully meet PLOS ONE’s publication criteria as it currently stands.

Some of the issues raised by the reviewers were not fully addressed. Therefore, we invite you to submit a revised version of the manuscript that addresses the remaining points.

We look forward to receiving your revised manuscript.

Kind regards,

Danillo G Augusto

Academic Editor

PLOS ONE

Reviewers' comments:

Reviewer's Responses to Questions

**Comments to the Author**

1. If the authors have adequately addressed your comments raised in a previous round of review and you feel that this manuscript is now acceptable for publication, you may indicate that here to bypass the “Comments to the Author” section, enter your conflict of interest statement in the “Confidential to Editor” section, and submit your "Accept" recommendation.

Reviewer #1: (No Response)

Reviewer #2: (No Response)

2. Is the manuscript technically sound, and do the data support the conclusions?

Reviewer #1: Yes

Reviewer #2: Partly

3. Has the statistical analysis been performed appropriately and rigorously? 

Reviewer #1: Yes

Reviewer #2: N/A

4. Have the authors made all data underlying the findings in their manuscript fully available?

Reviewer #1: Yes

Reviewer #2: Yes

5. Is the manuscript presented in an intelligible fashion and written in standard English?

Reviewer #1: Yes

Reviewer #2: Yes

6. Review Comments to the Author

Reviewer #1: The updated manuscript by Fidalgo, Torrezan and colleagues has been significantly improved. Clarification of the technical processing of the data aid the interpretability of the study.

For the comparison with non-cancer exomes I would still insist on performing a technical calibration of the sequencing data. Since case and control data was not jointly called technical differences between the variant calling pipelines could significantly bias the comparison. To prove the validity of this comparison, I would like to see the comparison of the synonymous variants between cases and controls. As you mentioned in your response letter, you have got the VCF file from controls with all variants. Allele frequencies comparison between cases and controls on synonymous variants could be performed with already available data relatively quickly. QQ-plot could be generated to evaluate the distribution of the test statistic.

Reviewer #2: The authors attempted to address my comments. They were unable to recruit additional family members and so were not able to collect additional information on the 5 families evaluated.

There is an error in line 92 of the revision. Part of the sentence appears to be missing.

The authors should check the manuscript for grammar errors.

7. PLOS authors have the option to publish the peer review history of their article (what does this mean?). If published, this will include your full peer review and any attached files.

Reviewer #1: No

Reviewer #2: No

---

## [Author Response · Author response to Decision Letter 1]

14 Sep 2021

To Prof. Dr. Danillo G Augusto

Academic Editor, PlosOne

 We are submitting a revised version of the manuscript “PONE-D-21-06922” for your reconsideration. The manuscript entitled “Family-based whole-exome sequencing identifies rare variants potentially related to cutaneous melanoma predisposition in Brazilian melanoma-prone families” was reviewed according to the remaining suggestions made by the reviewers. 

 Below we present the point-to-point response to each comment and suggestion. All authors agree with the final version of the manuscript.

 I hope you find it appropriate for publishing in PlosOne as a Research Article. 

Very best wishes,

Dirce Maria Carraro

Genomics and Molecular Biology Group, International Research Center

A.C. Camargo Cancer Center, R. Taguá, 440, Liberdade, São Paulo – SP, Brazil.

e-mail: dirce.carraro@accamargo.org.br

Reviewer #1: The updated manuscript by Fidalgo, Torrezan and colleagues has been significantly improved. Clarification of the technical processing of the data aid the interpretability of the study.

For the comparison with non-cancer exomes I would still insist on performing a technical calibration of the sequencing data. Since case and control data was not jointly called technical differences between the variant calling pipelines could significantly bias the comparison. To prove the validity of this comparison, I would like to see the comparison of the synonymous variants between cases and controls. As you mentioned in your response letter, you have got the VCF file from controls with all variants. Allele frequencies comparison between cases and controls on synonymous variants could be performed with already available data relatively quickly. QQ-plot could be generated to evaluate the distribution of the test statistic.

Response: We thank reviewer one for the first and second assessment of our manuscript and the important suggestions and commentaries. We were able to perform the comparison of the allele frequencies of synonymous variants between cases and controls. As it is can be seen in the graphics submitted in the PDF rebuttal letter (A: Density; B: Empirical cumulative distribution function), there is only a minor difference observed between cases and controls, which can be due to the different sequencing methods applied, different variant calling pipelines and different coverage mean, which was higher in controls than in patients. However, we believe this minor differences do not impact the purpose of using these non-cancer exome samples, as they were used only to exclude variants that could be common in the Brazilian population and absent in another genomic database, since at that time of our first data analysis and prioritization (performed in 2016) the ABraOM database was not available (database with exome sequencing data for 1,171 Brazilian healthy individuals, that was included in a second filter of our analysis). 

Reviewer #2: The authors attempted to address my comments. They were unable to recruit additional family members and so were not able to collect additional information on the 5 families evaluated.

There is an error in line 92 of the revision. Part of the sentence appears to be missing.

The authors should check the manuscript for grammar errors.

Response: We thank reviewer one for the first and second assessment of our manuscript and the important suggestions and commentaries. 

We apologized for the missing sentence. It was corrected to “The comparison between the number of variants called by each pipeline and the numbers of concordant calls are described in the supplementary table (S1).” (methods: line 92).

We have performed a detailed revision of the manuscript to correct grammar errors.

---

## [Decision Letter · Decision Letter 2]

13 Oct 2021

PONE-D-21-06922R2Family-based whole-exome sequencing identifies rare variants potentially related to cutaneous melanoma predisposition in  Brazilian melanoma-prone familiesPLOS ONE

Dear Dr. Carraro,

Thank you for submitting your manuscript to PLOS ONE. After careful consideration, we feel that it has merit but does not fully meet PLOS ONE’s publication criteria as it currently stands. Therefore, we invite you to submit a revised version of the manuscript that addresses the points raised during the review process.

We look forward to receiving your revised manuscript.

Kind regards,

Danillo G Augusto

Academic Editor

PLOS ONE

Journal Requirements:

Reviewers' comments:

Reviewer's Responses to Questions

**Comments to the Author**

1. If the authors have adequately addressed your comments raised in a previous round of review and you feel that this manuscript is now acceptable for publication, you may indicate that here to bypass the “Comments to the Author” section, enter your conflict of interest statement in the “Confidential to Editor” section, and submit your "Accept" recommendation.

Reviewer #1: (No Response)

Reviewer #2: All comments have been addressed

2. Is the manuscript technically sound, and do the data support the conclusions?

Reviewer #1: Yes

Reviewer #2: (No Response)

3. Has the statistical analysis been performed appropriately and rigorously? 

Reviewer #1: Yes

Reviewer #2: (No Response)

4. Have the authors made all data underlying the findings in their manuscript fully available?

Reviewer #1: Yes

Reviewer #2: (No Response)

5. Is the manuscript presented in an intelligible fashion and written in standard English?

Reviewer #1: Yes

Reviewer #2: (No Response)

6. Review Comments to the Author

Reviewer #1: The manuscript coherently outlines findings and analyses. However, I am still not convinced by the figures presented in the rebuttal letter. Since authors have allele frequencies in hand - there is a standard way of evaluating the technical bias in the data. Performing association study on synonymous variants and generating QQ-plot using obtained p-values. Genomic inflation (lambda) will become a numeric criterion for "matching" of the case and control data. I would like to kindly request this figure to be added to the supplement. Comments on the genomic inflation estimate (inflated/non-inflated) should be added to the main text.

Reviewer #2: (No Response)

7. PLOS authors have the option to publish the peer review history of their article (what does this mean?). If published, this will include your full peer review and any attached files.

Reviewer #1: No

Reviewer #2: No

---

## [Author Response · Author response to Decision Letter 2]

13 Dec 2021

To Prof. Dr. Danillo G Augusto

Academic Editor, PlosOne

 We are submitting the rebuttal letter for the third review of the manuscript “PONE-D-21-06922” entitled “Family-based whole-exome sequencing identifies rare variants potentially related to cutaneous melanoma predisposition in Brazilian melanoma-prone families” for your reconsideration. 

 First, we would like to apologize for the delay in submitting our response. Since the reviewer requested a novel analysis for which none of the authors of this study had sufficient expertise, we had to consult other experts in the field of population genetics. 

 Below we present the response to the comments of the reviewers. In previous revisions, the manuscript was reviewed according to all the suggestions made the reviewers. In this version, as justified below in our letter, we have only revised the reference list to ensure that it is complete and correct, as requested by the Journal. All authors agree with the final version of the manuscript.

 I hope you find it appropriate for publishing in PlosOne as a Research Article. 

Very best wishes,

Dirce Maria Carraro

Genomics and Molecular Biology Group, International Research Center

A.C. Camargo Cancer Center, R. Taguá, 440, Liberdade, São Paulo – SP, Brazil.

e-mail: dirce.carraro@accamargo.org.br

Reviewer #1: The manuscript coherently outlines findings and analyses. However, I am still not convinced by the figures presented in the rebuttal letter. Since authors have allele frequencies in hand - there is a standard way of evaluating the technical bias in the data. Performing association study on synonymous variants and generating QQ-plot using obtained p-values. Genomic inflation (lambda) will become a numeric criterion for "matching" of the case and control data. I would like to kindly request this figure to be added to the supplement. Comments on the genomic inflation estimate (inflated/non-inflated) should be added to the main text.

Response: We thank reviewer one for the assessment of our manuscript. We understand the remaining concerns of the reviewer regarding the control group and the request for performing genomic inflation estimate. However, we would respectfully like to argue that, unlike GWAS studies and other types of case-control studies that compare large populations and select candidate variants through a comparison of frequencies between cases and controls, our study has a different design. Our study is a family-based study, with the goal of identifying very rare genetic variants that are segregating in affected members of five families with high melanoma risk. In this scenario, the small number of control samples (10 Brazilian individuals) were used only to exclude variants that could be unique in the Brazilian population and absent in another genomic database, since at that time of our first data analysis and prioritization (performed in 2016) the ABraOM database was not available (database with exome sequencing data for 1,171 Brazilian healthy individuals, that was included in a second filter of our analysis). In this sense, it is of our understanding that since we did not used the controls to perform association analysis, given the small number of both controls (10 individuals) and patients (10 from 5 families), and the kinship relationship of the cases, comparing these groups with Q-Q plot analysis would not be appropriated.

Reviewer #2: All comments have been addressed.

Response: We thank reviewer one for the assessment of our manuscript and the important suggestions and commentaries in the previous rounds of revision.

---

## [Decision Letter · Decision Letter 3]

26 Dec 2021

Family-based whole-exome sequencing identifies rare variants potentially related to cutaneous melanoma predisposition in  Brazilian melanoma-prone families

PONE-D-21-06922R3

Dear Dr. Carraro,

We’re pleased to inform you that your manuscript has been judged scientifically suitable for publication and will be formally accepted for publication once it meets all outstanding technical requirements.

Kind regards,

Danillo G Augusto

Academic Editor

PLOS ONE

Additional Editor Comments (optional):

Reviewers' comments:

Reviewer's Responses to Questions

**Comments to the Author**

1. If the authors have adequately addressed your comments raised in a previous round of review and you feel that this manuscript is now acceptable for publication, you may indicate that here to bypass the “Comments to the Author” section, enter your conflict of interest statement in the “Confidential to Editor” section, and submit your "Accept" recommendation.

Reviewer #1: All comments have been addressed

2. Is the manuscript technically sound, and do the data support the conclusions?

Reviewer #1: Yes

3. Has the statistical analysis been performed appropriately and rigorously? 

Reviewer #1: Yes

4. Have the authors made all data underlying the findings in their manuscript fully available?

Reviewer #1: Yes

5. Is the manuscript presented in an intelligible fashion and written in standard English?

Reviewer #1: Yes

6. Review Comments to the Author

Reviewer #1: I am respectfully accepting your argument, though, as a suggestion, I would recommend that the brief statement about inability/irrelevance of genetic background matching is added to the manuscript methods/discussion section.

7. PLOS authors have the option to publish the peer review history of their article (what does this mean?). If published, this will include your full peer review and any attached files.

Reviewer #1: No

---

## [Editor Report · Acceptance letter]

17 Jan 2022

PONE-D-21-06922R3 

Family-based whole-exome sequencing identifies rare variants potentially related to cutaneous melanoma predisposition in  Brazilian melanoma-prone families 

Dear Dr. Carraro:

I'm pleased to inform you that your manuscript has been deemed suitable for publication in PLOS ONE. Congratulations! Your manuscript is now with our production department. 

Kind regards, 

on behalf of

Dr. Danillo G Augusto 

Academic Editor

PLOS ONE